# Implementing WHO guidance on conducting and analysing vaccination coverage cluster surveys: Two examples from Nigeria

John Ndegwa Wagai[1]☯*, Dale Rhoda[2]☯, Mary Prier[2‡], Mary Kay Trimmer[2‡], Caitlin B. Clary[2‡], Joseph Oteri[3], Bassey Okposen[3], Adeyemi Adeniran[4], Carolina Danovaro-Holliday[5]☯, Felicity Cutts[6]☯

1 Independent Consultant, Nairobi, Kenya, 2 Biostat Global Consulting, Worthington, OH, United States of America, 3 National Primary Health Care Development Agency, Abuja, Nigeria, 4 National Bureau of Statistics, Abuja, Nigeria, 5 World Health Organization, Geneva, Switzerland, 6 Department of Infectious Disease Epidemiology, London School of Hygiene and Tropical Medicine, London, United Kingdom

☯ These authors contributed equally to this work.
‡ These authors also contributed equally to this work
* johnwagai@gmail.com

**Data Availability Statement:** Data are held at the Nigeria Bureau of statistics microsite https://www.nigerianstat.gov.ng/nada/index.php/catalog/57 DDI

## Abstract

In 2015, the World Health Organization substantially revised its guidance for vaccination coverage cluster surveys (revisions were finalized in 2018) and has since developed a set of accompanying resources, including definitions for standardized coverage indicators and software (named the Vaccination Coverage Quality Indicators—VCQI) to calculate them.–The current WHO vaccination coverage survey manual was used to design and conduct two nationally representative vaccination coverage surveys in Nigeria–one to assess routine immunization and one to measure post-measles campaign coverage. The primary analysis for both surveys was conducted using VCQI. In this paper, we describe those surveys and highlight some of the analyses that are facilitated by the new resources. In addition to calculating coverage of each vaccine-dose by age group, VCQI analyses provide insight into several indicators of program quality such as crude coverage versus valid doses, vaccination timeliness, missed opportunities for simultaneous vaccination, and, where relevant, vaccination campaign coverage stratified by several parameters, including the number of previous doses received. The VCQI software furnishes several helpful ways to visualize survey results. We show that routine coverage of all vaccines is far below targets in Nigeria and especially low in northeast and northwest zones, which also have highest rates of dropout and missed opportunities for vaccination. Coverage in the 2017 measles campaign was higher and showed less geospatial variation than routine coverage. Nonetheless, substantial improvement in both routine program performance and campaign implementation will be needed to achieve disease control goals.

DOCUMENT ID: DDI-NGA-NBS-MICS5-2016-v01 and https://www.nigerianstat.gov.ng/nada/index.php/catalog/61 DDI DOCUMENT ID: DDI-NGA-NBS-PMCCS-2018-v01.

**Funding:** DAR and MLP and MKT are employed by Biostat Global Consulting a commercial firm and were funded via BMGF Investment ID 29065. FTC received consultancy fees from BMGF. CBC is funded on BMGF Investment ID 53009. MCDH is a WHO staff member. JNW was funded as the primary WHO consultant on the MICS/NICS and on the PMCCS. The content is solely the responsibility of the authors and does not necessarily represent the official views of the funding agencies. M. Carolina Danovaro[-Holliday] works for the World Health Organization. The author alone is responsible for the views expressed in this publication and they do not necessarily represent the decisions, policy or views of the World Health Organization, Biostat Global Consulting or BMGF. The funders did not play any role in the collection, analysis, interpretation, writing of final reports, or decision to submit this research.

**Competing interests:** DAR and MLP and MKT are employed by Biostat Global Consulting a commercial firm and received funds from BMGF Investment ID 29065. FTC received consultancy fees from BMGF. CBC is funded on BMGF Investment ID 53009. MCDH is a WHO staff member. JNW was funded as the primary WHO consultant on the MICS/NICS and on the PMCCS. The content is solely the responsibility of the authors and does not necessarily represent the official views of the funding agencies. M. Carolina Danovaro[-Holliday] works for the World Health Organization. The author alone is responsible for the views expressed in this publication and they do not necessarily represent the decisions, policy or views of the World Health Organization, Biostat Global Consulting or BMGF. The funders did not play any role in the collection, analysis, interpretation, writing of final reports, or decision to submit this Cover Letter research. The commercial affiliation of some of the authors does not alter our adherence to PLOS ONE policies on sharing data and materials.

# Background

Vaccination coverage–the proportion of the target population vaccinated with a given vaccine-dose–of the third dose of pentavalent vaccine (containing diphtheria-tetanus-pertussis (DTP), *H. influenzae* type b and hepatitis B vaccines), is used as a proxy indicator to monitor progress towards many global initiatives [1–3]. Household surveys are frequently conducted aiming to obtain more accurate information on vaccination performance than that derived from routine administrative reports, or to complement such reports [4–7]. Furthermore, Gavi-eligible countries are required to have done a nationally representative coverage survey (which may be multi-purpose such as the Demographic and Health Survey (DHS) or a UNICEF Multiple Indicator Cluster Survey (MICS)) within the last 5 years in order to apply for Gavi support as well as to conduct post campaign coverage surveys following any vaccination campaign (usually termed supplementary immunization activity (SIA)) supported by Gavi [8]. In 2015, the World Health Organization (WHO) updated its vaccination coverage survey guidance to promote the use of probability sampling with rigorous quality control, use of appropriate analysis and greater use of results to improve program performance (final version published in 2018) [9,10]. The guidance is supported by a set of materials including a list of standard questions and indicators, a tool called Vaccination Coverage Quality Indicators (VCQI) coded to calculate and tabulate most of these indicators [10–12], and intensive training through regional workshops and a large distance-based learning program in English and French, which is reaching several hundred participants around the world [13–16]. Briefly, VCQI is an open source collection of analytical programs, currently written in Stata for the analysis and visualisation of data collected from vaccination coverage surveys.

Challenges to implementing a high-quality household survey including selection bias, information bias, ascertainment of vaccination status and use of inappropriate analysis are well-known [4,17,18] but even when surveys are well-implemented, presentation of results may be hampered by a lack of standardized definitions of potential indicators and/or failure to report enough information to define those indicators clearly [10,19–21]. The Bill and Melinda Gates Foundation (BMGF); the United States Centers for Disease Control and Prevention (CDC); Gavi, the Vaccine Alliance; UNICEF, WHO, and independent experts therefore recently developed a white paper to help harmonize the collection and analysis of vaccination coverage data across the major survey programs [22].

In this paper, we present results from surveys conducted in Nigeria to illustrate many of the standard measures of vaccination coverage promoted in the White Paper for Routine Immunization (RI) and additional indicators for post campaign coverage surveys (PCCS) done after mass vaccination campaigns. We illustrate how moving beyond a single measure of coverage to an ensemble of quality indicators as produced by VCQI can help highlight priority areas for action.

# Materials and methods

## Data collection

We include data from two recent surveys that followed WHO-recommended procedures: the 2016–17 Multiple Indicator Cluster Survey/Nigeria Immunization Coverage Survey (MICS/NICS) and the 2018 National Post Measles Campaign Coverage Survey (PMCCS).

**MICS/NICS.** The 2016–17 MICS/NICS was a multi-purpose two-stage cluster sample survey which followed previously described methods [23–25]. The 2018 WHO coverage survey manual [9] recommends that before launching a survey only for vaccination, national immunization program (NIP) managers should determine whether another high-quality household

survey is being planned within the desired time frame and whether that sample would be appropriate for their needs. In 2015, the Government of Nigeria and UNICEF were planning a MICS hence it was decided to incorporate the NICS in the MICS [26]. The primary objective of NICS was to assess national and state levels of RI coverage for the "traditional vaccine doses"–one dose of BCG vaccine, three doses of DTP (in this instance, represented by pentavalent or "penta" vaccine), three doses of oral polio vaccine (OPV) and one dose of measles containing vaccine (MCV)–as well as yellow fever, hepatitis B birth dose and vitamin A supplements.

MICS/NICS was stratified by state (S1 Fig) with census enumeration areas (EAs) as the primary sampling units (PSUs). Sixty EAs were selected in each state (120 in Lagos and Kano) by simple random sample from the National Integrated Survey of Households round 2 (NISH2) master sample, based on a list of EAs prepared for the 2006 Population Census. In each sampled EA, after listing all households, 16 were selected by systematic random sampling. For NICS, to enable estimation of vaccination coverage among the smaller cohort of children aged 12–23 months with precision for the 3rd dose of pentavalent vaccine (Penta3) no wider than +/- 10% in each state, an additional 10–30 EAs were selected in 20 states. Details of sampling are in [24 Appendix A]. The overall MICS/NICS duration was about a year and half with about four months for data collection.

Standard MICS questionnaires were administered with an additional set of questions on reasons for no vaccination; data were collected using computer assisted personal interviewing (CAPI). In the supplemental clusters, only the modules on household characteristics and vaccination were administered. Respondents were asked if they had a home-based record (HBR) of vaccination and if this was available, the dates of each vaccination were transcribed. Then, for children not fully vaccinated, an additional question was asked to ascertain and record if any other vaccines had been received that were not shown on the card. If no HBR was seen, caretakers were asked whether the child had received each vaccine-dose in the schedule and their answer was recorded as verbal recall. The availability of HBR in Nigeria has been below 50% in all recent surveys and even though recent guidance recommends that interview teams might visit health facilities to seek documentation for children lacking HBRs [4,9,22], health facility visits were not included in this survey due to logistical and resource constraints.

**PMCCS.** The PMCCS was conducted between January and April 2018, following measles SIAs targeting children aged 9 to 59 months held from October to December 2017 in northern Nigeria and February to March 2018 for Southern States [27]. A forthcoming special supplement to the journal *Vaccine* describes numerous aspects of these SIAs. The primary objective of this PMCCS was to estimate the coverage of measles vaccination during the SIA in each state, in the Federal Capital Territory (FCT), and nationally. Some specific Local Government Areas (LGAs) in Borno and Adamawa states were excluded from the sampling frame due to security concerns. Results from those states should be interpreted in light of the exclusions.

The survey methods followed the 2015 draft WHO guidelines [28] (finalised in 2018 without substantial change in content) [9,10] and data collection was done using CAPI. A stratified two-stage cluster sampling design was used, as in MICS/NICS. After household listing, households were selected centrally using simple random sampling without replacement from the list of households with eligible children aged 9 to 59 months. Assuming an expected SIA coverage of 90%, half-width confidence interval around state-level estimates of 8% (i.e., 90% +/- 8% coverage estimate) with an alpha level (type I error) of 5%, the effective sample size (i.e., sample size per stratum under a simple random sampling assumption) was n = 101 [9, Annex B1]. This was increased to 210 households with eligible children to account for the cluster survey design effect and expected non-response rate [9, Annex B1]. Seven households with eligible children were therefore randomly selected from each of 30 EAs in every state and the FCT. A

short questionnaire was administered to record where the child lived during the SIA; knowledge of and source of information about the SIA; vaccination during the SIA according to SIA card, finger-mark or recall; any adverse events following immunization; reasons for not attending the SIA if relevant; and whether the child had received measles vaccination before the SIA according to HBR or recall.

Both MICS/NICS and PMCCS were implemented by the Nigeria Bureau of Statistics (NBS) which ensured that data were collected in line with Nigeria's laws and ethical requirements. In both surveys, only adults who give verbal consent were interviewed and permission for secondary analysis was obtained from NBS using anonymised datasets where all personal identifying information had been stripped.

The maps in this manuscript were made by the authors using Stata Version 16 (Stata-Corp. 2019. Stata Statistical Software: Release 16. College Station, TX: StataCorp LLC.) using shapefiles downloaded from https://energydata.info/dataset/nigeria-administrative-boundaries-2017/resource/2342db86-823e-4bf2-920f-58be34968f36 which are licensed under Creative Commons Attribution 4.0. The shapefiles have not been previously copyrighted to our knowledge. The boundaries and names shown and the designations used on maps in this manuscript do not imply the expression of any opinion whatsoever on the part of the authors or of the World Health Organization concerning the legal status of any country, territory, city or area nor of its authorities, or concerning the delimitation of its frontiers or boundaries.

### Data analysis

**Weighting and post stratification.**    The MICS/NICS used a MICS spreadsheet template to calculate cluster-specific survey weights. Base weights were calculated as the inverse of the multi-stage probability of selection. These were adjusted for non-response at the household and child levels. The weights were not post-stratified; that is, the state-level sums of weights were not scaled to match totals or ratios from an administrative list [29]. The weights were scaled to sum to the total nationwide MICS/NICS sample size of children aged 12–23 months.

PMCCS weights were computed by adjusting observations with inverse selection probabilities of EAs and households, adjusting for non-response at the EA, household and child levels and finally post stratifying the weights with the estimated number of children aged between 9 and 59 months in each state obtained from the campaign micro plan [30].

For reporting of results, sample weights were applied to outcomes where all respondents are in the denominator and for which results will be generalized to the entire eligible population. Analyses were un-weighted if the denominator included only a subset of respondents.

**Indicator definitions and reporting recommendations.**    S1 and S2 Tables summarise the definitions of some of the main indicators calculated by VCQI; details are available in the VCQI documentation [31–33]. WHO recommends that RI coverage be tabulated according to the source of information: HBR; health facility-based record (FBR) (if available); caretaker recall only and HBR+FBR+recall [22]. Coverage should also be classified as crude (including evidence of vaccination at any age) or valid (children who have documented dates of each vaccine-dose and received each dose according to WHO guidance on minimum ages and intervals between doses) [34].

In the early years of the WHO Expanded Programme on Immunization (EPI), only 4 basic vaccines against six diseases (BCG, DTP (3 doses), OPV (3 doses) and MCV) were included in national immunization schedules of low and middle-income countries and a "fully vaccinated" child had received all these doses. The number of vaccines included in schedules now varies greatly between countries hence a fully vaccinated child can be defined as a child who has received all doses of the 4 "basic vaccines" (adapting the earlier EPI definition to replace DTP

with pentavalent vaccine if used) or a more stringent definition of a child who has received all the vaccines in the country's vaccination schedule in the relevant time period for the cohorts in the study. For simplicity we use only the first definition here.

Missed Opportunities for Vaccination (MOV) due to non-simultaneous vaccination were defined as documented vaccination visits where a child received one or more vaccines but not all the vaccines for which they were eligible. For example, a visit in which an MCV dose was given but OPV or Penta vaccines were not given even though those 3-dose series were not completed and the minimum interval had passed since their last dose. This approach was endorsed by WHO's Immunization and vaccines related implementation research advisory committee (IVIR-AC) in 2016 [35].

Dropout was defined as the percentage of children 12–23 months who received a dose in a multi-dose sequence but failed to receive a subsequent or the final dose in the sequence. This is typically calculated between first and third doses of pentavalent vaccines as (penta1-penta3/penta1) expressed in percentage.

**Analysis.**   Both surveys were analysed with WHO's VCQI software using Stata version 15 [11,33,36].

The recent WHO white paper [22] encourages special attention and clear documentation of how the following issues are handled during data cleaning and indicator construction. S4 Table details the following:

- Definitions of eligible population and denominators for each indicator

- Handling evidence from tick marks

- Handling imperfect date values

- Handling missing values or 'unsure' or 'do not know'

- Steps to differentiate RI doses from SIA doses

- Definitions of valid doses

- Calculation of confidence intervals

- How many decimal places to report

The analysis plan for each survey aligned well with that described in the WHO white paper [22]. In VCQI, results tables and figures are presented in stratum order that can be customised by the user. In the Nigeria example, we show national, zonal and state results in the same figure so that results for each state can easily be compared to the average in its zone and the national average. National and zonal names have been capitalised and horizonal lines have been used to demarcate different zones. These style and formatting options can be changed by the user according to their needs.

## Results

### Description of the survey samples

In MICS/NICS, 2,810 EAs were selected into the sample. Interviews were conducted in 2,702 EAs; 108 EAs were excluded for reasons of security and inaccessibility [27 Table HH.3] Interviews were successfully conducted in 40,518 (98.7%) of 42,981 households selected into the sample. In those households, caretakers had responsibility for 6,360 children aged 12–23 months. Interviews were completed for 6,268, yielding a vaccination-specific caretaker-response rate of 98.6%.

In the PMCCS, 2,239 of 2,340 EAs selected in the first stage of sampling were listed and covered during the fieldwork period while 101 EAs were inaccessible due to insecurity especially in Borno, Yobe and Adamawa states. Of the 7,090 households selected, interviews were successfully conducted in 6,819, a household response rate of 96.2%.

**Primary indicators: Weighted and applicable to the population -routine vaccination coverage among children aged 12–23 months, MICS/NICS 2016–17.** Although caregivers of 51.3% (95% CI 49.0, 53.6) of children reported that their child had ever received a HBR or card, these were seen for only 29.0% (95% CI 27.3, 30.8) of children, ranging from as low as 5% in Sokoto state to a high of 68% in Lagos (Fig 1). HBRs were seen more frequently in urban than rural areas and in wealthier households (data not tabulated). In Fig 1, the vertical bar in each 2-D distribution represents the survey point estimate. The base of each distribution spans the 2-sided 95% confidence interval. The height of the shape indicates the relative degree of confidence that coverage falls at that value. The shapes consist of scaled stacked confidence intervals with the 95% interval at the base and a 1% interval at the peak. See [9, Annex M].

Table 1 shows crude vaccination coverage of each vaccine-dose according to source of information, and valid coverage by the time of the survey. At the national level, crude coverage was low for all vaccines (e.g. 48.7% (95% CI: 46.4, 51.1) for Penta1 and 41.7% (39.5,43.9) for MCV). Only 22.9% (21.2, 24.6) of children had received all basic vaccines even accepting maternal recall for children without HBR. Remarkably, 40% (37.5, 42.1) of children had received none of the basic vaccinations (Table 1), ranging from 60.5% (57.1, 63.9) in the northwest to only 7.5% (5.1, 11.0) in the southeast.

Coverage varied widely by state (Figs 2 and S2) [37]. Penta1 coverage, commonly taken as an indicator of access to vaccination services, was over 80% in the three southern zones but very low in northwest and north-eastern zones (Fig 2). For Penta3 and MCV, however, coverage was well below targets even in southern states.

Vaccination coverage was virtually identical in boys and girls. More than twice as many children had never been vaccinated in rural (49%) than urban (20%) areas. Coverage was lower among poorer and less educated families and certain ethnic groups [25].

Just over half of caretakers reported obtaining vaccination at government health centres, while only 5% used private facilities (12% in the south-eastern zone and among mothers with higher education)–data not tabulated. Mobile or outreach clinics were mentioned by only 8%, with a high of 12% in north central zone. By contrast, campaigns or SIAs were the reported source of vaccination for 23% of children (higher in the northwest (34%) than southwest (12%)).

**Post-campaign measles vaccination coverage among children aged 9–59 months, PMCCS 2018.** Only 3% of children were not resident during the time of the SIA [27 Table 2.1a] and only 4% of respondents had not received information about the SIA (though in 2 states, over 10% lacked information) [27 Table 2.2a]. Direct communication from community health workers, town criers and community leaders was by far the commonest source of information [27 Table 2.2a]. Overall, 87.5% (95% CI: 86.2, 88.7) of children were estimated to have received MCV during the SIA; approximately half showed the SIA card while 17% had finger-mark evidence, the remainder being verbal recall [27 Table 2.5b]. Importantly, the SIA was relatively effective in reaching previously unreached children (82.4%), although coverage was higher (91.6%) among children who had previously received MCV (Table 2). Nationally, 11.2 percent of all children aged between 9 and 59 months remained unvaccinated against measles following the campaign [27 Table 2.8a]. Coverage varied by state (Fig 3) [37]. In Sokoto, Zamfara and Yobe states where MCV coverage was <17% in MICS/NICS (S2 Fig), over 85% coverage was reached by the SIA [27 Table 2.5a]. Coverage also varied within states–organ pipe plots of the proportion of children vaccinated in each cluster showed which EAs

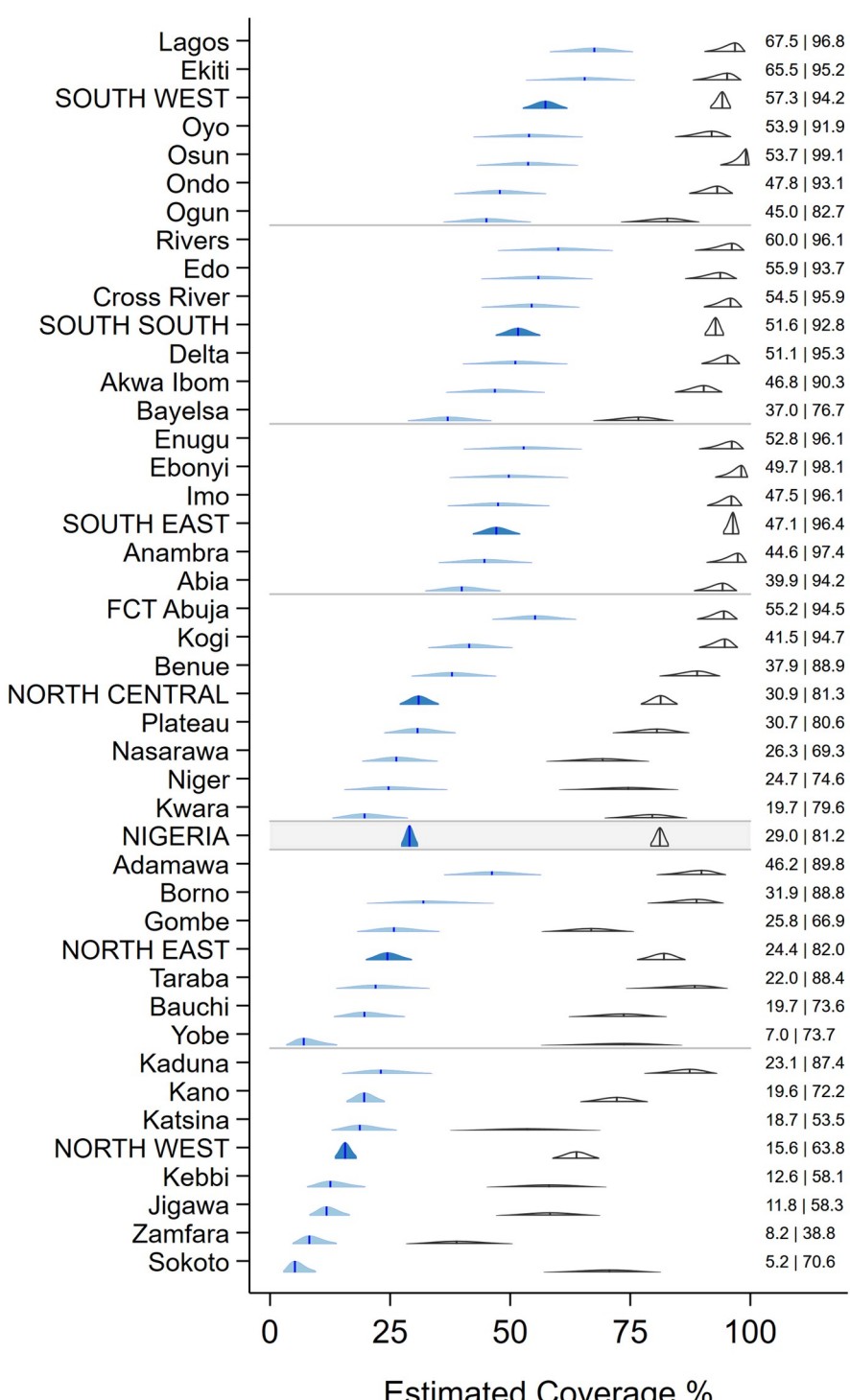

Text at right: Point Estimate | Point Estimate

Colored shape is card availability; gray hollow shape is card ever received

**Fig 1. Weighted percentage of children aged 12–23 months who were reported to have ever received a card and whose card was seen during the survey, Nigeria MICS/NICS 2016–17.**

**Table 1. Weighted percentage crude (ignoring age at vaccination or interval between doses) and valid (respecting the schedule) vaccination coverage for the basic EPI vaccines, children aged 12–23 months, MICS/NICS 2016–17.**

| | Crude coverage | | | | | | | Valid coverage by HBR; denominator is: | | |
| | by HBR† | | by Recall | | by HBR + Recall | | children with HBR‡ | all children | | |
| | % | 95% CI | % | 95% CI | % | 95% CI | % | % | 95% CI |
|---|---|---|---|---|---|---|---|---|---|
| BCG | 28.0 | (26.4, 29.7) | 25.3 | (23.3, 27.3) | 53.5 | (51.1, 55.9) | 87.9 | 25.6 | (24.0, 27.2) |
| HBV0 | 20.5 | (19.0, 22.0) | 9.7 | (8.9, 10.7) | 30.2 | (28.4, 32.0) | 36.2 | 10.6 | (9.5, 11.9) |
| OPV0 | 24.0 | (22.4, 25.6) | 23.5 | (21.7, 25.3) | 47.4 | (45.4, 49.5) | 41.4 | 12.1 | (10.9, 13.3) |
| OPV1 | 27.0 | (25.3, 28.7) | 22.8 | (21.0, 24.7) | 49.7 | (47.7, 51.7) | 85.2 | 24.1 | (22.5, 25.7) |
| OPV2 | 24.3 | (22.7, 25.9) | 18.3 | (16.9, 19.8) | 42.5 | (40.6, 44.4) | 72.9 | 20.3 | (19.0, 21.8) |
| OPV3 | 21.7 | (20.2, 23.2) | 11.6 | (10.3, 13.0) | 33.2 | (31.5, 35.0) | 46.4 | 12.9 | (11.8, 14.0) |
| Penta1 | 27.7 | (26.1, 29.4) | 21.2 | (19.2, 23.3) | 48.7 | (46.4, 51.1) | 88.0 | 25.1 | (23.5, 26.7) |
| Penta2 | 25.2 | (23.7, 26.9) | 14.7 | (12.9, 16.7) | 39.9 | (37.8, 42.1) | 78.2 | 22.0 | (20.5, 23.6) |
| Penta3 | 23.0 | (21.4, 24.5) | 10.4 | (9.0, 11.9) | 33.3 | (31.4, 35.3) | 51.6 | 14.2 | (13.1, 15.4) |
| MCV | 20.5 | (19.1, 22.0) | 21.2 | (19.4, 23.1) | 41.7 | (39.5, 43.9) | 53.2 | 15.0 | (13.8, 16.4) |
| YF | 19.6 | (18.2, 21.0) | 19.2 | (17.6, 20.9) | 38.8 | (36.7, 40.9) | 52.6 | 14.8 | (13.6, 16.1) |
| Fully vaccinated* | | | | | 22.9 | (21.2, 24.6) | 30.8 | 8.2 | (7.4, 9.2) |
| No vaccinations** | | | | | 39.8 | (37.5, 42.1) | 6.1 | 72.5 | (70.8, 74.2) |
| Unweighted sample size | 2,084 | | 4,184 | | 6,268 | | 2,084 | 6,268 | |
| Weighted sample size | 1,883 | | 4,386 | | 6,268 | | 1,883 | 6,268 | |

NOTES.

Abbreviations: HBR = home-based record (or vaccination card); BCG = Bacillus Calmette–Guérin; HBV = hepatitis B vaccine; OPV = oral polio vaccine;

penta = pentavalent DTP-Hib-HepB vaccine; MCV = measles-containing vaccine; YF = yellow fever vaccine. The number next to the vaccine indicates the dose, 0 means that is a birth dose. Inactivated polio vaccine (IPV), which was introduced in Nigeria in 2015, is not included here.

95 percent confidence intervals reported in parentheses.

* Fully vaccinated = the percentage of children aged 12–23 months who received all of the "basic vaccines", i.e. 1 dose of BCG vaccine, 3 doses of DTP-containing vaccine ("Penta"), 3 doses of OPV and 1 dose of MCV before the survey.

** No vaccinations = the percentage of children aged 12–23 months who had received none of the "basic vaccines".

† The percentage of children aged 12–23 months for whom a home-based record (HBR) was available and reviewed for evidence of vaccination was 29.0% (95% CI: 27.3,30.8).

‡ By convention, estimates where the denominator is a subset of respondents, are unweighted and presented without a confidence interval.

had markedly low numbers of children vaccinated (S3 Fig) [9 Section 6.1.2] [38–40]. There was no sex difference in measles vaccination during the SIA or by urban and rural residence [27 Table 2.5b]. Infants aged between 9 and 11 months had lower coverage during the campaign (75.5%, 95% CI: 67.2, 82.3) compared to older children (84–90% in each older year of age) [27 Table 2.5b].

**Secondary indicators of vaccination quality from MICS/NICS.** *Timeliness of vaccination.* Among children whose HBR was seen, a high proportion received "non-valid" vaccine doses before the recommended age or with too short an interval between doses. Overall, 13% received Penta1 before age 6 weeks and 8% received subsequent doses of pentavalent vaccine after too short an interval (<28 days), while 17% received MCV before the recommended age of 9 months (273 days) [25 Tables IM.15 and IM.16]. Zones and states with lower crude coverage also tended to have more early doses– 29% of children with a record of MCV vaccination were vaccinated before 9 months in the North West and 25% in North East zone (Fig 4).

As recommended in the WHO White Paper on surveys [22], Fig 5 shows vaccine coverage plots by age of child for select doses for the 2,017 children with a HBR. The curves reveal some doses being administered early and a large portion administered quite late with many children

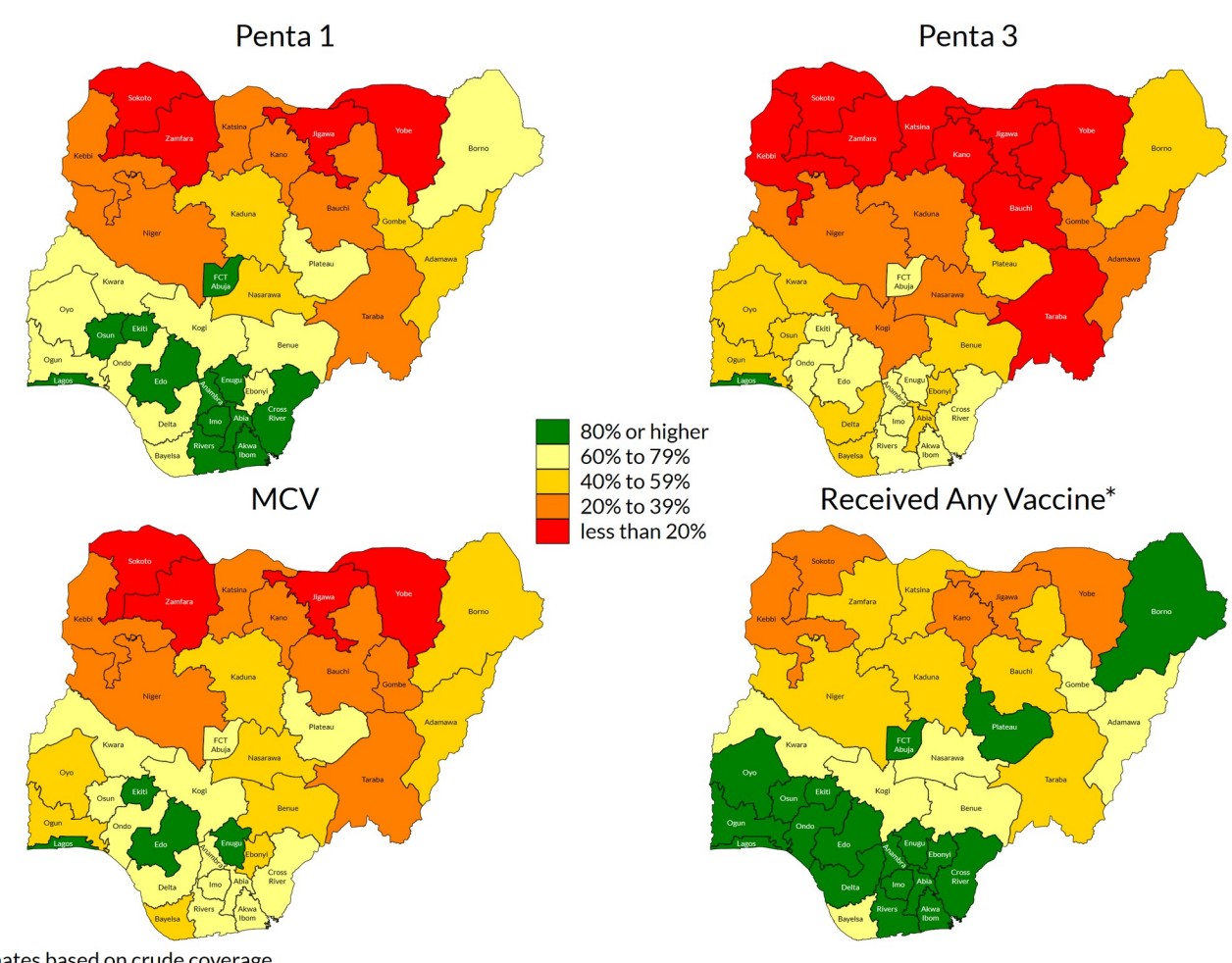

Estimates based on crude coverage
*Received any of the doses in the fully vaccinated dose list

**Fig 2. Weighted crude coverage of first and third dose of pentavalent vaccine, of measles vaccine and of any of the basic vaccines, children aged 12–23 months, by state, Nigeria MICS/NICS 2016–17.**

receiving the second and third doses of OPV or Penta after 6 months of age (180 days). S4–S6 Figs show coverage curves for MCV in Nigeria, the North West zone, and Lagos, respectively. Lagos had the highest proportion of children with HBRs (67.5%) (Fig 1). The North West zone had the highest proportion of early MCV doses among children with HBRs (29.3%) (Fig 4).

**Table 2. Proportion of children aged 9 months to 59 months who received measles vaccine during the measles campaign, Nigeria PMCCS 2018.**

|  | Vaccinated during SIA N = 8,884 | | Not vaccinated during SIA N = 1,269 | | N | Weighted N |
|---|---|---|---|---|---|---|
|  | % | 95% CI | % | 95% CI |  |  |
| Nigeria | 87.5 | (86.2, 88.7) | 12.5 | (11.3, 13.8) | 10,153 | 35,939,548 |
| MCV vaccination status before the campaign | | | | | | |
| Had received ≥1 dose of MCV before the campaign | 91.6 | (90.2, 92.8) | 8.4 | (7.2, 9.8) | 5,567 | 19,706,044 |
| Had never received MCV before the campaign | 82.4 | (80.1, 84.4) | 17.6 | (15.6, 19.9) | 4,586 | 16,233,504 |

Abbreviations: CI = Confidence Interval.

The results in this table are from weighted analysis and the CI calculation considers the sampling design & weights.

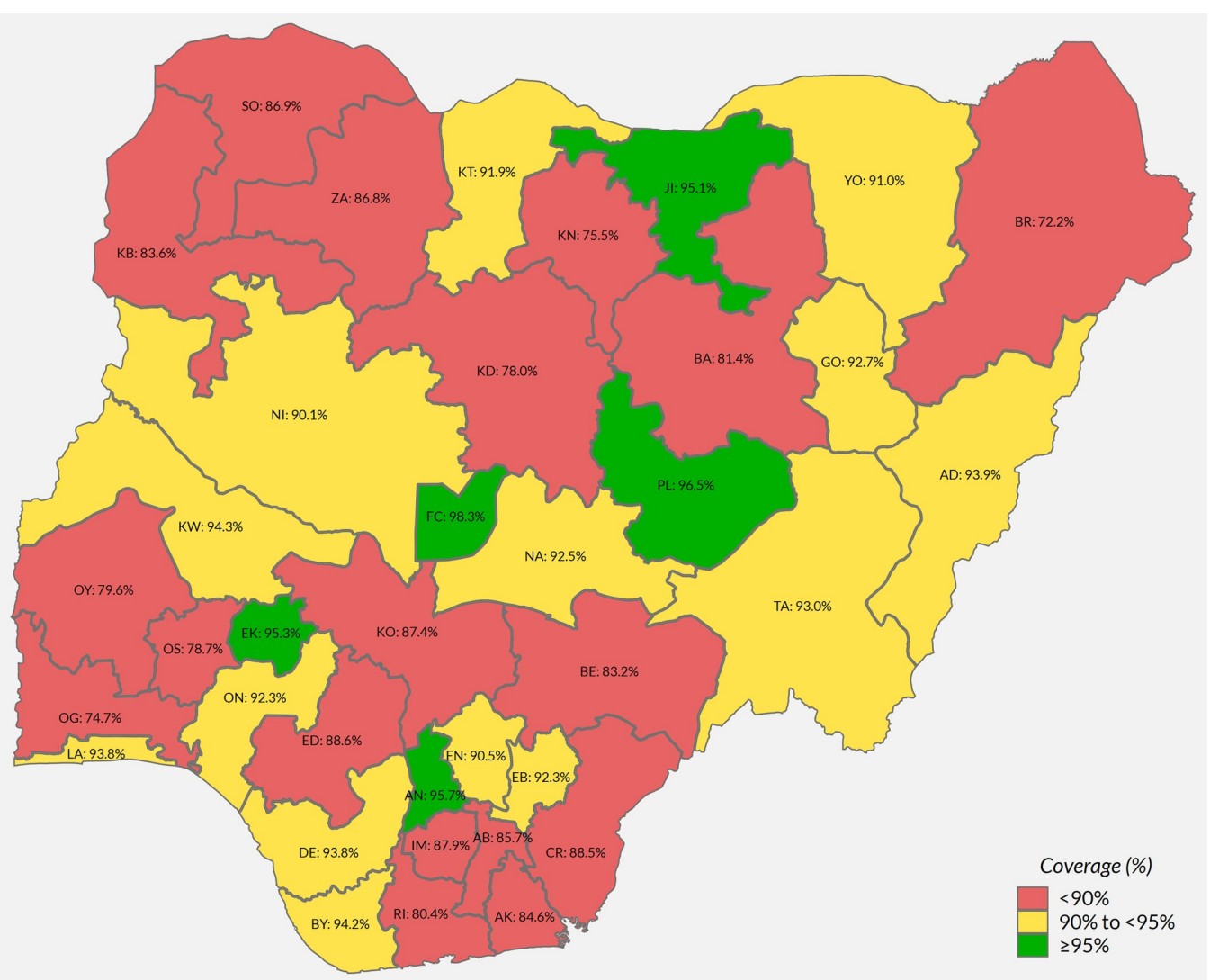

**Fig 3. Weighted percentage of children 9 to 59 months vaccinated during the 2017–18 measles SIA, by state.**

All three figures reveal an appreciable number of doses given early. S7 Fig shows coverage curves by age for Penta1-3 in Lagos and show vaccine administration that is much closer to the recommended age than the national curves in Fig 5.

**Dropout.** Fig 6 shows very high dropout especially in states in the North West and North East zones, where fewer than half the children who received Penta1 completed the three-dose series. Of note, dropout was substantially higher when using information from caretaker recall than among children with a HBR (Table 1 and S3 Fig).

**Missed Opportunities for Vaccination (MOVs).** Of 1,912 children aged 12–23 months who had at least one documented age-eligible vaccination visit in their HBR, 1,005 (53%) experienced at least one MOV for any of the 4-basic vaccine/doses or yellow fever (Fig 7). Of these, 45% had all MOVs later corrected (i.e. they received the missed vaccines at a later date) (S8 Fig) and 17% had some corrected, while 38% never received any of the vaccines that had been missed (not tabulated here). MOVs were high in all zones and states. Among states where more than 50 children showed HBRs, the percent of children with one or more MOVs ranged

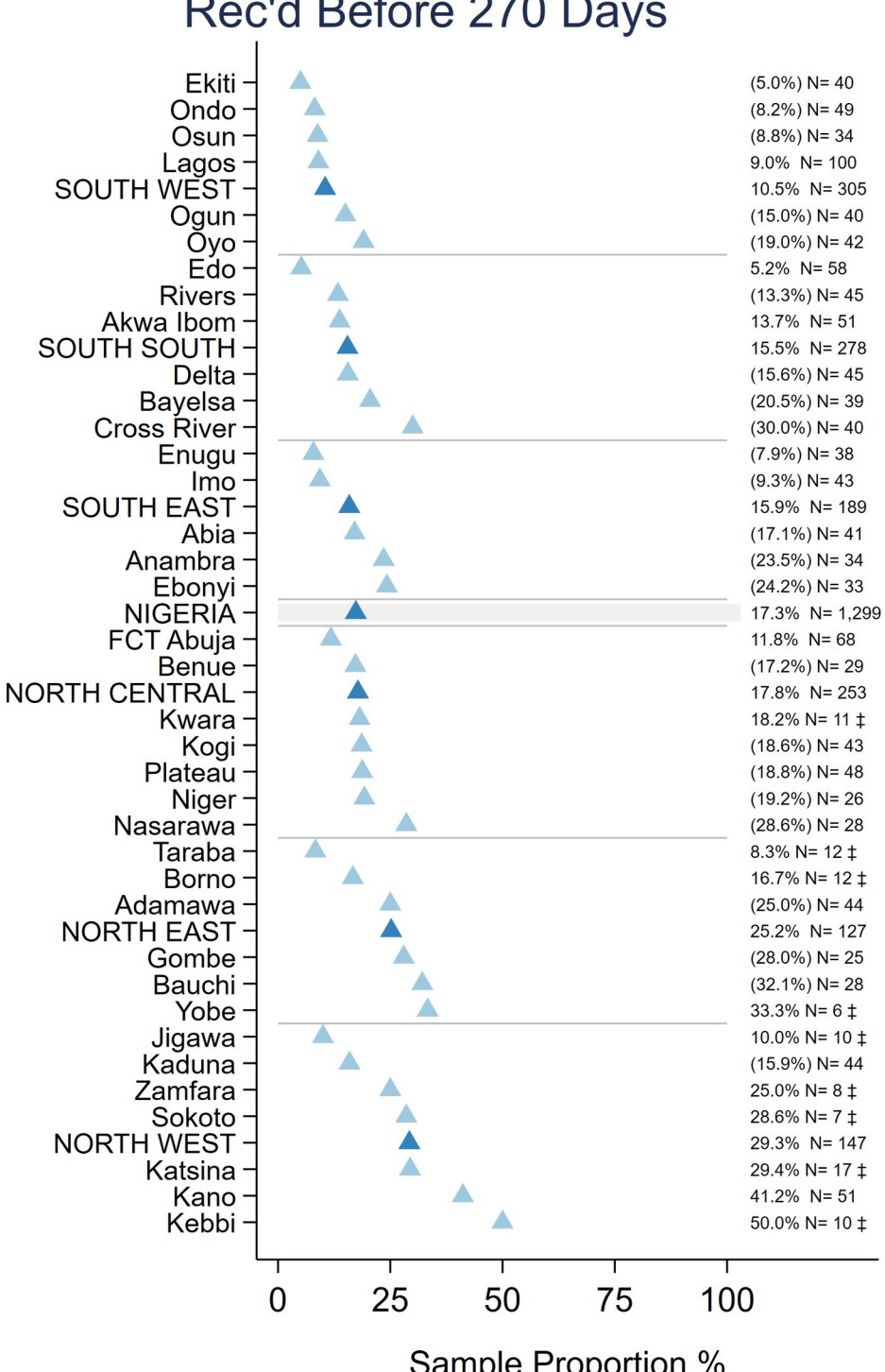

**Fig 4. Unweighted percentage of children aged 12–23 months with a date of vaccination for MCV who received the vaccine before age 270 days, by state, Nigeria MICS/NICS 2016–17.**

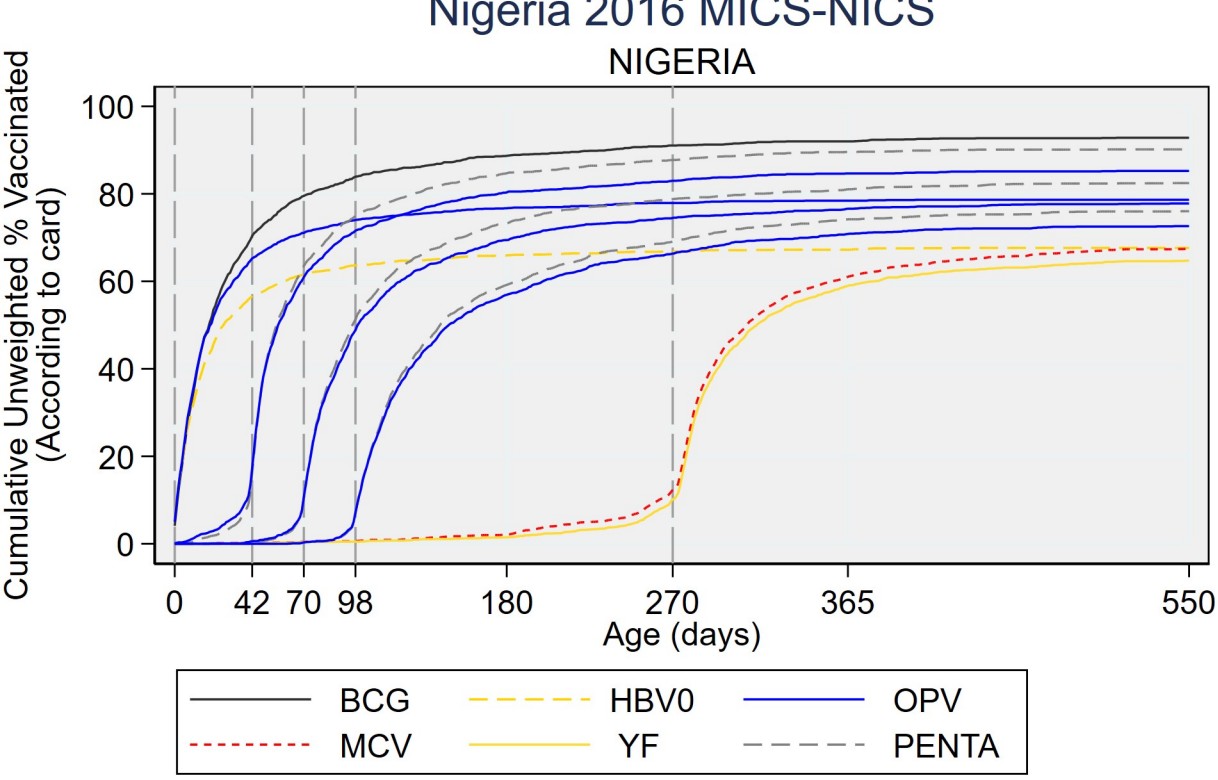

Vertical dashed lines mark scheduled vaccination ages: 0, 42, 70, 98 & 270 days.
Denominator is number of respondents with cards with dates (n=2,017).

**Fig 5. Cumulative coverage percentages for selected doses, by age of child, among children who showed a HBR to the interviewer, Nigeria MICS/NICS 2016–17.**

from 34% in Enugu to 75% in Kano (Fig 7). Even in the South East which had the highest over-all coverage, 41% of children with HBRs had at least one MOV for these vaccines (Figs 7 and 8) [41].

The proportion of MOVs that was later corrected was much higher in southern than north-ern states (Figs 8 and S8). Elapsed time from the first MOV until the dose was later received is represented in Fig 9 using cumulative distributions, in days. Red lines mark the 50[th] percentile, indicating that in most cells, 50% of the corrections occurred within one or two months of the missed opportunity.

Lagos and Kano were the states with largest MICS/NICS survey samples, 120 clusters each, because of detailed sub-state reporting requirements. MOVs were much more prevalent in Kano than Lagos with the proportion of Kano's children experiencing MOVs ranging from 10% (for Penta2) to 44% (for OPV1) (Fig 10). Kano was the only state with more than 50 chil-dren with HBRs who had MOVs; N = 73. Of those, only one-fourth had all MOVs corrected before the survey (Figs 10 and S8). The supplement shows the prevalence of MOVs for select doses in every state (S9 and S10 Figs).

VCQI calculates several other indicators which we do not have room to discuss in detail: proportion of dose intervals that were too short or very long; proportion of children who would have received a valid dose if there had been no MOVs, proportion of vaccination visits that produce 1+ MOVs, and others [32,33].

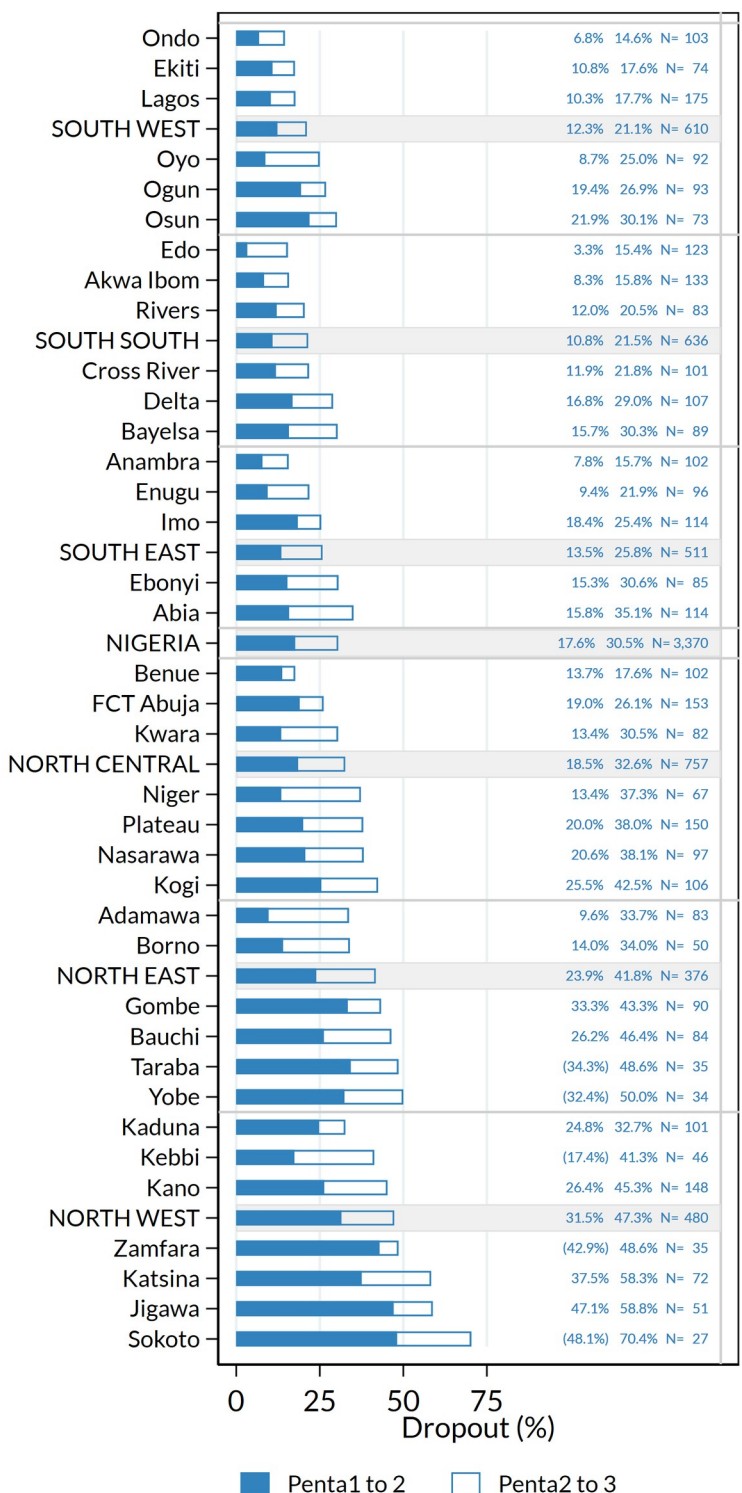

**Fig 6. Percentage dropout according to HBR or recall between first and second doses (Penta1-2) and between first and third doses (Penta1-3) of pentavalent vaccine among children who received Penta1, by state, Nigeria MICS/ NICS 2016–17.**

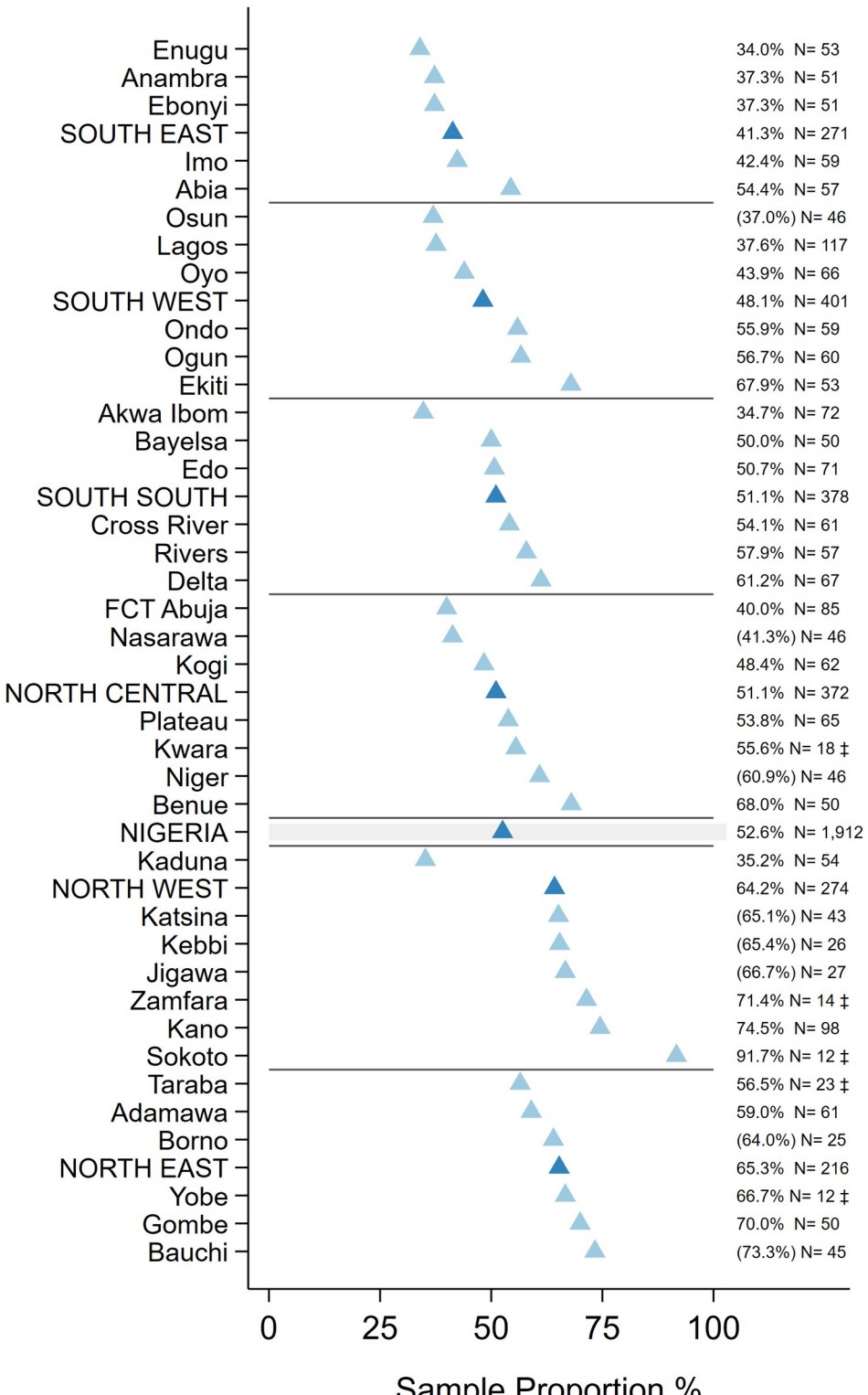

Text at right: Unweighted sample proportion (%) and N
Parentheses () mean 25 ≤ N < 50 and ‡ means N < 25.

**Fig 7. Unweighted percentage of children aged 12–23 months with a HBR who had at least one MOV for BCG, HBV0, OPV0-3, Penta1-3, MCV or YF due to non-simultaneous vaccination, by state, Nigeria MICS/NICS 2016–17.**

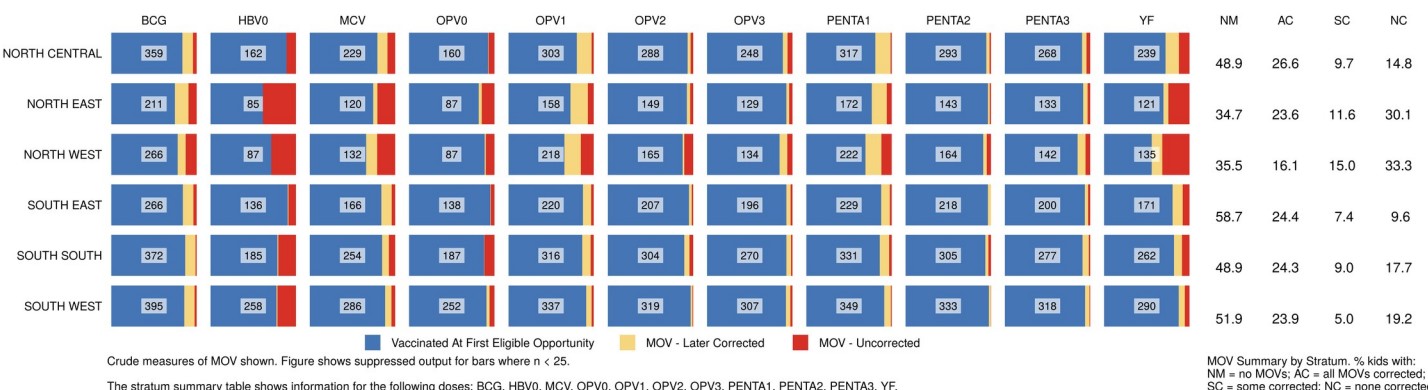

**Fig 8. Percentage of children vaccinated at the first eligible opportunity, and percent who experienced one or more missed opportunities, whether later corrected or not.** The numbers in the centre of each cell portray the number of children in that zone (row) who had 1+ vaccination visits when age eligible to receive that dose (column). Nigeria MICS/NICS 2016–17.

## Discussion

Since the early years of the EPI, household surveys have been promoted to monitor vaccination coverage [42]. With the aim of obtaining results that are as accurate, precise, and reliable as possible, countries are currently encouraged to commission institutions or partners with statistical and survey expertise to conduct high-quality and statistically sound independently implemented vaccination coverage surveys. The reasons for this include the increasing complexity and cost of EPI with the addition of new vaccines targeting different age groups; increasing coverage levels in most countries, which calls for more precise coverage estimates; improved survey and statistical methods as well as tools to manage large databases; and a world where accountability is key for governments, partners supporting EPI and for the beneficiaries of the immunization programme [9]. During the development and rollout of updated survey guidance, WHO and partners noted the need to improve the standardization of coverage indicator definitions, survey questionnaires and the analysis and presentation of results [4,10,21,43]. In this paper we have illustrated the use of a standardized, freely available tool to analyse coverage surveys and the presentation of results on indicators that are harmonized across the major survey programs which monitor vaccination coverage [22]. The suite of indicators presented here, along with others detailed in VCQI documentation provides managers with information to guide improvements to their programs [11].

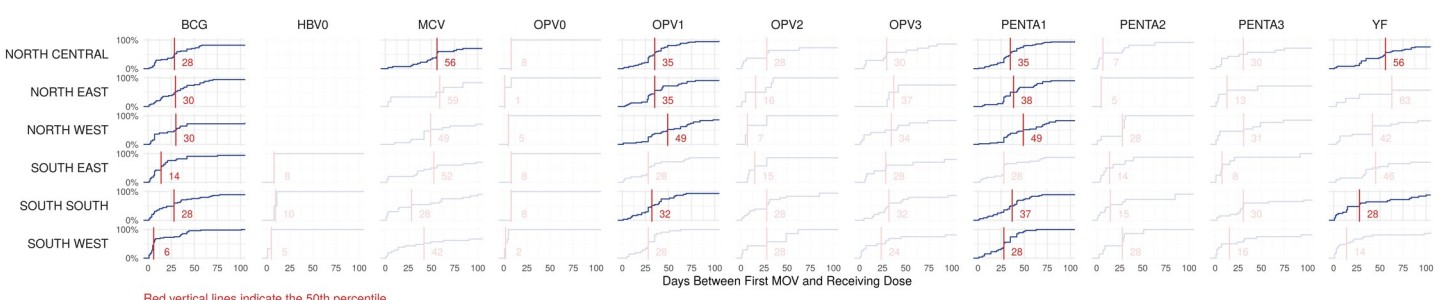

**Fig 9. Cumulative distributions of time to correction, in days, for missed opportunities for vaccination, by dose and zone, Nigeria MICS/NICS 2016–17.**

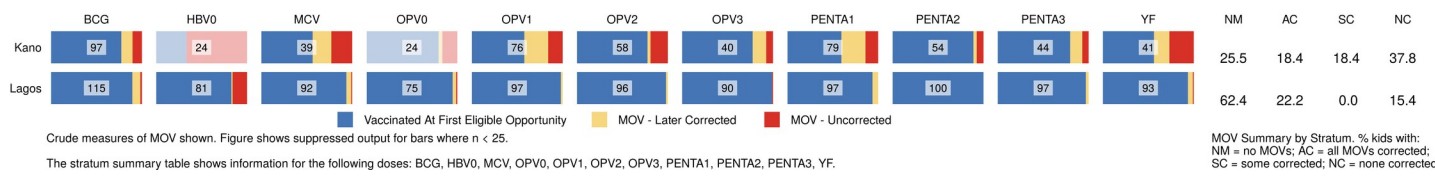

**Fig 10. Missed opportunities for vaccination in Kano and Lagos, Nigeria MICS/NICS 2016–17.**

The 2016–17 MICS/NICS showed several findings of importance to the Nigeria vaccination program. Coverage of all vaccines was low even when using the most liberal definition of crude coverage by the time of the survey including maternal recall. Only just over half of respondents said the child had ever received a HBR, and only 29% could show the HBR to interviewers. Survey results for crude coverage therefore rely heavily on caretaker recall, which reduces data reliability [44–46] especially for the number of doses received in a multi-dose series–studies suggest that in some settings caretakers tend to under-report the number of doses received [44,46].

Coverage varied greatly between states and, though not presented here, according to background factors including education and wealth [25]. Low coverage of Penta1 is often taken to indicate low access to vaccination services [9] and many northern states had crude coverage for Penta1 below 40%, contrasting starkly with the African regional average for 2016 of 84% [47]. There was a gradient of coverage for all vaccines from lowest in the north, especially those in the North West and North East zones, to higher in the south but only one state (Lagos) achieved over 80% crude coverage of Penta3. To compound the problem of low average coverage, Nigeria was recently ranked highest among 45 Gavi-supported countries for inequity in coverage [48,49].

Crude coverage of MCV was higher than that of Penta3, and four states achieved over 80% coverage for measles. This probably reflects the inclusion of campaign doses in the measure of measles vaccination coverage [50], as about half the age cohort included in MICS-NICS would have been eligible for the 2015–16 measles SIA. This conclusion is supported by the finding that MCV coverage was slightly lower than Penta3 coverage among children with cards (where documented doses reflect RI) whereas it was about twice as high as Penta3 by recall (Table 1). The low availability of HBR meant that most information was obtained by verbal recall, though some missed-opportunities to give Penta when a child presented late for vaccination cannot be excluded. Since MCV is administered via many of the same fixed facilities and outreach posts in campaigns as in routine services, it would be difficult for respondents to differentiate between RI and campaign doses several months or years after a campaign took place. The WHO white paper encourages further operational research on how to frame questions to elicit this type of information [22]. Notably, the cumulative coverage plots by age showed that there was very little increase in MCV coverage after 12 months of age. Current WHO recommendations are for routine vaccinations to be offered to children over 12 months of age who have not yet received all recommended doses [51] and it is important for this "catch-up" policy to be implemented in Nigeria.

The 2018 PMCCS confirmed that SIAs reach much higher coverage than routine in much of Nigeria and importantly, SIA coverage among previously MCV zero-dose children was high [52], suggesting that lack of acceptance of vaccination is not a major barrier. A mixed-methods study in two Nigerian states found that receipt of MCV was related to awareness of vaccination, parental education, maternal participation in decision-making, presence of a government vaccination facility, and lack of barriers such as having to pay for vaccination [53]. The intensive information campaigns leading up to SIAs combined with making vaccination sites more

accessible likely both contribute to the SIA's success [54]. Nonetheless, SIA coverage was below that required for measles elimination and in some survey clusters fewer than 25% of children in the coverage survey had received the SIA dose. Although cluster-level results are not statistically significant in a typical household survey, this imprecise suggestion of geospatial variation in SIA coverage was confirmed in separate formal analyses that were possible because geographic coordinates were obtained in the Nigeria surveys [52]. While geospatial differences in coverage were much smaller for doses given in SIAs than in RI, SIA implementation needs further improvement as well as greatly strengthening RI to approach Nigeria's measles elimination goal [27].

Indicators of the quality of vaccination services highlight multiple problems in all states. Although low card availability in MICS/NICS limits the generalisability of measures of vaccination quality, there is no reason to expect *better* performance among the children whose HBR were not seen. Many doses were given before the recommended age or with too short an interval between doses. Valid coverage was therefore even lower than crude coverage by HBR, the difference being most marked for OPV3, Penta3, MCV and YF (Table 1). For MCV, although 225 (17.3%) of 1,299 children whose age at vaccination could be calculated received the dose before age 9 months, most of the early doses were received between 6–8 months of age (and over half in their 8th month) when immunogenicity is only moderately reduced [55]. In Nigeria, the frequent conduct of SIAs should mean that most children have an opportunity for additional doses to ensure protection.

Dropout between vaccine doses in the primary series was high, especially in states with low coverage of Penta1. Under the 4th strategic objective of Global Vaccine Action Plan Monitoring and Evaluation/Accountability Framework, measuring of the dropout rates between first and third dose of DTP-containing vaccines is a key component of tracking the functionality of the health system. Incomplete vaccination in Nigeria has been linked to parental belief that the series was complete; inconvenient vaccination sites; lack of awareness and absence of vaccinators or of vaccine [56,57]. Further health-facility-based studies would be useful to determine how to address problems of health worker absence, health worker attitudes, and other potential barriers to completion of the series [43]. Given the big difference in dropout measured by card versus recall in MICS/NICS, health facility-based studies would also have the advantage of enabling the use of register data to obtain more accurate information on dropout at the health facility level.

MOVs due to non-simultaneous vaccination were common. This is particularly worrisome where access is poor–when a mother does bring a child for vaccination it is especially important to ensure that all indicated vaccines are administered. In the North East and North West zones, most children with a MOV never received the missed vaccine, and no zone compensated for more than 60% of the missed opportunities, hence the overall effect on coverage was large (Fig 8 and S8–S10 Figs). As noted in the WHO coverage survey reference manual, it is possible that some children had a valid contraindication to receipt of a vaccine, but these are likely to be a small minority given the rarity of true contraindications [58,59]. While health system barriers could account for lower vaccination coverage and MOV observed especially in North East and North West Zones, health-facility based surveys of MOVs need to be conducted to investigate the causes of MOV and assess MOVs among children attending primarily for reasons other than vaccination and a national plan of action to reduce MOVs developed and acted on [60]. A review of MOV literature from 1992–2014 concluded that lack of standardized analyses was a barrier to assessing MOV trends over time [61]. The updated WHO survey manual and VCQI address this issue with clear MOV-related indicators that may be used to examine MOVs in both past and current coverage survey datasets [9,33].

Few countries have lower routine vaccination coverage or greater geographical inequity than Nigeria [62]. Since 2000, coverage gains in Nigeria have mainly been limited to the south, with

most of the northwest and northeast stagnating or deteriorating [62]. In summer of 2017 the National Primary Healthcare Development Agency declared a state of emergency on routine immunization in Nigeria [63,64]. A national coordination centre was established to provide guidance and resources to states [65,66], several of which have signed memoranda of understanding with national and international partners to strengthen routine immunization programs [67,68]. Improving access to routine vaccination e.g. by increasing the frequency and reliability of fixed and outreach services is likely to be an important strategy to increase coverage. In addition to improving access, much can be achieved by improving the standard of services offered at existing facilities and information given to parents, so that children who do access vaccination complete the vaccination series on time. Future facility-based surveys should complement household surveys for the needed improvement of health worker performance.

Our results are subject to many limitations, mainly related to the surveys themselves, including the outdated sampling frame, areas that could not be visited, limited and heterogeneous availability of HBRs, and no visits to health facilities. Exclusion of insecure areas and lack of records of previous SIA doses might have biased the findings regarding measles-zero-dose children in the PMCCS [69]. However, the main findings stand, coverage is low, dropouts are high, campaigns reached many people who were previously measles-zero dose, but not at the same level as for those who had already a previous MCV dose, and VCQI can help rapidly run these analysis and visuals.

Household surveys are done regularly in Nigeria, as in many low-income countries, and standardising their analysis and reporting of results will help to increase their use for programmatic action. WHO is rolling out training on the conduct of high-quality immunization surveys and how to use VCQI for analysis. Immunization program managers and their partners should become familiar with new guidelines and available tools and support for high quality surveys, definitions and outputs of a broad array of indicators, and consider other indicators that they may wish to have calculated (e.g. factors associated with receipt of vaccination). A useful starting point is to request secondary analyses of earlier household surveys that included vaccination coverage in their country, to identify possible strengths and shortcomings and to provide information to help plan the design and analysis of upcoming surveys. Lastly, those commissioning and conducting surveys should ensure that microdata are made publicly available, so the most can be made of survey results to plan improvements to their program.

## Supporting information

**S1 Fig. Map of Nigeria, showing zones (coloured) and states.**
(TIF)

**S2 Fig. Crude coverage with one dose of measles-containing vaccine (MCV1), children aged 12–23 months, Nigeria MICS/NICS 2016–17, by state and zone.**
(TIF)

**S3 Fig. Post-measles campaign coverage survey (PMCCS) "organ pipe plots" of percentage of children in each survey cluster who had received a measles vaccine SIA dose (by card or history) for four states, Nigeria PMCCS 2018.**
(TIF)

**S4 Fig. Cumulative measles-containing vaccine (MCV) coverage plot by age of child (days) in Nigeria among children aged 12–23 months who showed a home-based record or card, MICS/NICS 2016–17.**
(TIF)

**S5 Fig. Cumulative measles -containing vaccine (MCV) coverage plot by age of child (days) in North West zone, among children aged 12–23 months who showed a home-based record or card, Nigeria MICS/NICS 2016–17.**
(TIF)

**S6 Fig. Cumulative measles -containing vaccine (MCV) coverage plot by age of child (days) in Lagos state, among children aged 12–23 months who showed a home-based record or card, Nigeria MICS/NICS 2016–17.**
(TIF)

**S7 Fig. Cumulative pentavalent dose 1, 2 and 3 vaccine coverage plot by age of child (days) among children aged 12–23 months who showed a home-based record or card, Lagos state, Nigeria MICS/NICS 2016–17.**
(TIF)

**S8 Fig. Unweighted percentage of children aged 12–23 months with a home-based record (HBR) and at least one missed opportunity for vaccination (MOV) for BCG, HBV0, OPV0-3, Penta1-3, MCV or YF who later received all the vaccines that had been missed ("corrected" MOVs), by state and zone, Nigeria MICS/NICS 2016–17.**
(TIF)

**S9 Fig. Proportion of children with a home-based record (HBR) who were vaccinated at the first eligible opportunity, and proportion who experienced one or more missed opportunity for vaccination (MOV), whether later corrected or not, by state for Northern states, Nigeria MICS/NICS 2016–17.**
(TIF)

**S10 Fig. Proportion of children with a home-based record (HBR) who were vaccinated at the first eligible opportunity, and proportion who experienced one or more missed-opportunity for vaccination (MOV), whether later corrected or not, by state for Southern states, Nigeria MICS/NICS 2016–17.**
(TIF)

**S1 Table. Selected definitions used in the Vaccination Coverage Quality Indicators (VCQI) tool for indicators calculated using weighted analyses.**
(DOCX)

**S2 Table. Selected definitions used in VCQI for indicators calculated using unweighted analyses which represent a subset of the total target population.**
(DOCX)

**S3 Table. Dropout rates between different vaccine-dose combinations in the vaccination series, by source of information, Nigeria MICS/NICS 2016–17.**
(DOCX)

**S4 Table. Data cleaning details for Nigeria MICS/NICS 2016–17 and PMCCS 2018.**
(DOCX)

## Acknowledgments

The MICS/NICS survey was commissioned by the National Primary Healthcare Development Agency (NPHCDA) and implemented by the National Bureau of Statistics (NBS) with technical support and funding from WHO, UNICEF and the Bill and Melinda Gates Foundation (BMGF) and technical support from Biostat Global Consulting.

The PMCCS was commissioned by the NPHCDA and implemented by the NBS between January and April 2018 with funding from BMGF, Gavi, the Vaccine Alliance, and the Government of Nigeria and with technical assistance from WHO, the Centre for Disease Control and Prevention—National Stop Transmission of Polio (CDC-NSTOP) programme, and the United Nations Children's Fund (UNICEF).

Specific mention goes to Dr Faisal Shuaib, Dr Eric Nwaze, Ms Ijeoma Onuoha, of NPHCDA; Dr Isiaka Olarewaju, Mr Biyi Fafunmi, Mr Salihu Isaac, Mr Kola Ogundiya, Mr Bolakale Kareem, Mr Shamsudeen Lawal, Ms Abiola Adeleke of NBS; Dr Tove Ryman, Jenny Sequeira and Yusuf Yusufari of BMGF; Dr David Brown of Brown Consulting Group International, Dr Wenfeng Gong, Dr Kyla Hayford of the International Vaccine Access Center, Johns Hopkins Bloomberg School of Public Health; Dr Modibo Kassogue, Denis Jobin, Denis Businge of UNICEF Nigeria Country Office; Bo Robert Beshanski-Pedersen of UNICEF Headquarters; Dr Fiona Braka, Dr Rachael Seruwange, Dr Anne Jean Baptiste, Dr Daniel Ali of WHO, Nigeria Country Office for their contributions to the planning, coordination, implementation and dissemination of the MICS/NICS and PMCCS surveys.

**Additional supporting materials**

The 2018 Vaccination Coverage Cluster Survey Reference Manual is available in English and French [9,70] and is supported not only by the Vaccination Coverage Quality Indicators (VCQI) Stata programs [11] and the white paper [22] mentioned in this manuscript, but also by a freely-available sample size calculator [71] and a so-called *cookbook* consisting of 1–2 page summaries of the steps to conduct a coverage survey [72]. Figs 8–10 and S9 and S10 were made with an interactive R Shiny tool that any VCQI user may use to browse their own output; the tool can export images or tables or datasets to document the prevalence of missed opportunities for vaccination [41]. WHO welcomes feedback on all of these materials; address correspondence to M. Carolina Danovaro at danovaroc@who.int and Dale Rhoda at Dale.Rhoda@biostatglobal.com.

## Author Contributions

**Conceptualization:** John Ndegwa Wagai, Dale Rhoda, Mary Prier, Joseph Oteri, Bassey Okposen, Adeyemi Adeniran, Carolina Danovaro-Holliday, Felicity Cutts.

**Data curation:** John Ndegwa Wagai, Dale Rhoda, Mary Prier, Mary Kay Trimmer, Caitlin B. Clary, Joseph Oteri, Adeyemi Adeniran, Carolina Danovaro-Holliday, Felicity Cutts.

**Formal analysis:** John Ndegwa Wagai, Caitlin B. Clary, Adeyemi Adeniran, Carolina Danovaro-Holliday, Felicity Cutts.

**Investigation:** Dale Rhoda, Joseph Oteri, Bassey Okposen, Carolina Danovaro-Holliday, Felicity Cutts.

**Methodology:** John Ndegwa Wagai, Dale Rhoda, Mary Kay Trimmer, Carolina Danovaro-Holliday, Felicity Cutts.

**Project administration:** Bassey Okposen.

**Resources:** John Ndegwa Wagai, Carolina Danovaro-Holliday.

**Software:** Dale Rhoda, Caitlin B. Clary.

**Supervision:** John Ndegwa Wagai, Carolina Danovaro-Holliday.

**Validation:** Mary Prier, Mary Kay Trimmer, Joseph Oteri, Bassey Okposen, Carolina Danovaro-Holliday, Felicity Cutts.

**Visualization:** John Ndegwa Wagai, Mary Prier, Mary Kay Trimmer.

**Writing – original draft:** John Ndegwa Wagai, Carolina Danovaro-Holliday, Felicity Cutts.

**Writing – review & editing:** John Ndegwa Wagai, Dale Rhoda, Mary Prier, Mary Kay Trimmer, Caitlin B. Clary, Joseph Oteri, Bassey Okposen, Adeyemi Adeniran, Carolina Danovaro-Holliday, Felicity Cutts.

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
