## [Decision Letter · Decision Letter 0]

30 Sep 2020

PONE-D-20-18456

Implementing WHO guidance on conducting and analysing vaccination coverage cluster surveys: Two examples from Nigeria

PLOS ONE

Dear Dr. Wagai,

Thank you for submitting your manuscript to PLOS ONE. After careful consideration, we feel that it has merit but does not fully meet PLOS ONE’s publication criteria as it currently stands. Therefore, we invite you to submit a revised version of the manuscript that addresses the points raised during the review process.

We look forward to receiving your revised manuscript.

Kind regards,

Olalekan Uthman, MD, MPH, PhD, FRSPH, FHEA

Academic Editor

PLOS ONE

Journal Requirements:

'DAR and MLP and MKT were funded via BMGF Investment ID 29065.  FTC received consultancy fees from BMGF.  CBC is funded on BMGF Investment ID 53009.  MCDH is a WHO staff member. JNW was funded as the primary WHO consultant on the MICS/NICS and on the PMCCS.  

The content is solely the responsibility of the authors and does not necessarily represent the official views of the funding agencies. M. Carolina Danovaro[-Holliday] works for the World Health Organization. The author alone is responsible for the views expressed in this publication and they do not necessarily represent the decisions, policy or views of the World Health Organization. The funders did not play any role in the collection, analysis, interpretation, writing of final reports, or decision to submit this research.

https://www.gatesfoundation.org'

We note that one or more of the authors are employed by a commercial company: Independent Consultant, and Biostat Global Consulting .

3. We note that Figures in your submission contain map images which may be copyrighted.

a. You may seek permission from the original copyright holder of Figures to publish the content specifically under the CC BY 4.0 license. 

4. Please note that in order to use the direct billing option the corresponding author must be affiliated with the chosen institute. Please either amend your manuscript or remove this option (via Edit Submission).

5. Please remove your figures from within your manuscript file, leaving only the individual TIFF/EPS image files, uploaded separately.  These will be automatically included in the reviewers’ PDF.

6. Please include captions for your Supporting Information files at the end of your manuscript, and update any in-text citations to match accordingly. Please see our Supporting Information guidelines for more information: http://journals.plos.org/plosone/s/supporting-information

Reviewers' comments:

Reviewer's Responses to Questions

**Comments to the Author**

1. Is the manuscript technically sound, and do the data support the conclusions?

Reviewer #1: Yes

2. Has the statistical analysis been performed appropriately and rigorously? 

Reviewer #1: Yes

3. Have the authors made all data underlying the findings in their manuscript fully available?

Reviewer #1: Yes

4. Is the manuscript presented in an intelligible fashion and written in standard English?

Reviewer #1: Yes

5. Review Comments to the Author

Reviewer #1: Dear Authors,

This is an interesting topic and properly researched work, however, there are some concerns. Please find below my comments:

Abstract

1. The abstract did not highlight the findings of the surveys, the differences and importance of using either of the two.

2. What is the purpose of this study?

3. Highlight priority areas for action.

Background

1. Line 40: Meaning of DTP?

2. Be consistent with the use of GAVI or Gavi.

3. Line 51: The references should be: (9-10)

4. Brief description of VCQI is needed.

5. What are the specific challenges of implementing a high-quality household survey?

6. Line 60: Please identify the key partners.

Methods

1. Ethical considerations of the initial data collection process for the two surveys were not mentioned.

2. Line 117: Please write the full meaning of SIA.

Results

1. There is need to define and differentiate state data from regional data when reporting. Sokoto and Lagos are states while North West or North East are regions. It is confusing for someone who is not conversant with Nigeria’s geopolitical structure to understand the differences without proper definition of these variables.

2. Lines 244 – 252: It will be nice having tables to showcase this statements and possibly univariate regression to show if there are any significant differences.

Discussion

1. Line 470: MOV varies across the country, what are the likely reasons for these differences.

2. Recall bias is also a major limitation.

3. Line 488: Please re-phrase the sentence.

Figures and tables

1. Most of the figure and table titles are rather too long.

2. Figures 1, 4, 6 and 7: The states and regions should be clearly defined and separated. The state, regional and national data were mixed up.

6. PLOS authors have the option to publish the peer review history of their article (what does this mean?). If published, this will include your full peer review and any attached files.

Reviewer #1: No

<gdiv></gdiv>

---

## [Author Response · Author response to Decision Letter 0]

3 Dec 2020

A point by point response to the Academic editor and reviewer has been included in the "Repose to Reviewers" letter.

---

## [Decision Letter · Decision Letter 1]

8 Feb 2021

Implementing WHO guidance on conducting and analysing vaccination coverage cluster surveys: Two examples from Nigeria

PONE-D-20-18456R1

Dear Dr. Wagai,

We’re pleased to inform you that your manuscript has been judged scientifically suitable for publication and will be formally accepted for publication once it meets all outstanding technical requirements.

Kind regards,

Olalekan Uthman, MD, MPH, PhD, FRSPH, FHEA

Academic Editor

PLOS ONE

Reviewers' comments:

Reviewer's Responses to Questions

**Comments to the Author**

1. If the authors have adequately addressed your comments raised in a previous round of review and you feel that this manuscript is now acceptable for publication, you may indicate that here to bypass the “Comments to the Author” section, enter your conflict of interest statement in the “Confidential to Editor” section, and submit your "Accept" recommendation.

Reviewer #1: All comments have been addressed

2. Is the manuscript technically sound, and do the data support the conclusions?

Reviewer #1: Yes

3. Has the statistical analysis been performed appropriately and rigorously? 

Reviewer #1: Yes

4. Have the authors made all data underlying the findings in their manuscript fully available?

Reviewer #1: Yes

5. Is the manuscript presented in an intelligible fashion and written in standard English?

Reviewer #1: Yes

6. Review Comments to the Author

Reviewer #1: This is a good and satisfactory revision. The reviewers concerns were addressed and the current product is okay for publication.

7. PLOS authors have the option to publish the peer review history of their article (what does this mean?). If published, this will include your full peer review and any attached files.

Reviewer #1: No

---

## [Editor Report · Acceptance letter]

16 Feb 2021

PONE-D-20-18456R1 

Implementing WHO guidance on conducting and analysing vaccination coverage cluster surveys: Two examples from Nigeria 

Dear Dr. Wagai:

I'm pleased to inform you that your manuscript has been deemed suitable for publication in PLOS ONE. Congratulations! Your manuscript is now with our production department. 

Kind regards, 

on behalf of

Dr. Olalekan Uthman 

Academic Editor

PLOS ONE